



**STABILITY OF SOIL ORGANIC MATTER IN CRYOSOLS OF MARITIME**
**ANTARCTIC: INSIGHTS FROM 13-C NMR AND ELECTRON SPIN RESONANCE**
**SPECTROSCOPY**
**Evgeny Abakumov, Ivan Alekseev**
*Department of Applied Ecology, Saint-Petersburg State University, 199178, 16-line 2, Vasilyevskiy*
*Island, Russian*
*Key words*
Antarctica, soil organic matter, stabilization, humic acids
*Key points*
Investigation of Antarctic soil organic matter stability
Humic acids of superficial horizons contain more aromatic carbon
Humic acids of isolated layers contain more free radicals
**Abstract**
In this study, the soil organic matter (SOM) was analyzed from different sample areas (surface
level and partially isolated supra-permafrost layer) of the tundra-barren landscape of the Fildes
Peninsula, King George Island, Western Antarctica. We found that the humic acids (HAs) of the
cryoturbated, buried areas had lower amounts of alkylaromatic and protonized aromatic
compounds. In contrast, the HAs from the surface layers contain less alkyl carbon components.
The free radical content was higher in the surface layers than in the buried layers due to the
presence of fresh organic remnants in superficial soil samples. New data on SOM quality from
these two representative Cryosols will enable more precise assessment of SOM stabilization rate
in sub-Antarctic tundras. Comparison of the $^{13}$C-NMR spectra of the HAs and the bulk SOM
revealed that humification occurs in the Antarctic and results in accumulation of aromatic and
carboxylic compounds and reductions in alkylic ones.

*Highlights*
Investigation of Antarctic soil organic matter stability
Humic acids of superficial horizons contain more aromatic carbon
Humic acids of isolated layers contain more free radicals
**1. Introduction**




Polar soils play a key role in global carbon circulation and stabilization as they contain
maximum stocks of soil organic matter (SOM) within the whole pedosphere (Schuur et al, 2015).
Cold climate and continuous and discontinuous permafrost result in the stabilization of essential
amounts of organic matter in soils, biosediments, and grounds of the polar biome (Zubrzycki et al,
44  2014).

Global climate changes and permafrost degradation have led to the exposure of huge
pools of organic matter to microbial degradation (Schuur et al, 2015) and other environmental
risks. Polar SOM represents a vulnerable carbon source, susceptible to remobilization under
increasing temperatures  (Schuur et al, 2015, Ejarque, Abakumov, 2016). In order to better
understand the implications of permafrost SOM for greenhouse gas emissions, accurate
knowledge of its spatial distribution, both in terms of quantity and quality (e.g. biodegradability,
chemical composition, and humification stage) is needed in addition to effective evaluation of
SOM's temporal dynamics  (Fritz et al, 2014, Vasilevitch et al, 2018 ).
Current estimations of soil organic carbon (SOC) stocks are around 1307 Pg throughout
the northern circumpolar region (Hugelius et al, 2014). These amounts surpass previous
estimates (Tarnocai et al, 2009) and grossly exceed the total carbon contained in the world's
vegetation biomass (460 - 650 Pg) or in the atmosphere (589 Pg) (Tarnocai et al, 2009).
However, the aforementioned SOM/SOC stock estimations are still poorly constrained (Hugelius
et al, 2014). This uncertainty is largely caused by the estimates having been calculated from
observations that are highly spatially clustered (Hugelius et al, 2014) while extensive land areas
remain uncharacterized due to the logistic difficulties of reaching these sites. Additionally, the
calculation of these stocks are based on estimated data on soil bulk density and carbon values
derived from dichromate oxidation methods (Abakumov, Popov, 2005, Polyakov et al, 2017).
The stocks of SOM in the Antarctic are underestimated compared to the Arctic because
of the lack of the data for many parts of this continent, due to the high content of stones in the
soils and the high variability in the carbon content of the fine earth. Stocks of organic carbon in
the Antarctic soil have been reported as 0.5 kg/m$^2$ in its polar deserts, about 1.0 kg/m$^2$ in its
barrens, up to 3 - 5 kg/m$^2$ in the sub-Antarctic tundra, and up to 30 kg/m$^2$ in the penguin
rockeries of the maritime islands (Abakumov, 2010, Abakumov, Mukhametova, 2014,
Abakumov et al, 2016).
Stability and biodegradability are the key features of SOM that should be taken into
account when estimating current and future carbon stocks and organic matter quality and
dynamics. Stability is related to humification degree, as more advanced stages in the
humification process involve depletion of the labile molecules, as well as an increase in the bulk
aromaticity, which confers higher stability to the SOM. A number of proxies have been used to
trace humification rate and SOM stability, including aromaticity level (Vasilevitch et al, 2018,
Kniker, 2007). Also the ratio of C-Alkyl : C-Aryl and C-Alkyl : O-N-alkyl have been
successfully used to assess humification degree (Kinker, 2007). C/H ratio from humic acids
(HAs) has been used as an index of molecular complexity, as more degrees of conjugation imply
less hydrogenation of the carbon chains  (Zaccone et al, 2007) and C/N has been used as a
measure of histic material degradation (Lodigin et al, 2014). $^{13}$C-NMR spectrometry provides
information on the diversity in carbon functional structures (carbon species) and has been used to
evaluate changes in SOM during decomposition and humification. More specifically, high
phenolic (150 ppm), carboxyl-C (175 ppm) and alkyl-C (30 ppm) groups, combined with low O-
alkyl carbons, have been associated with advanced humification stages (Zech et al, 1997). So far,
studies of SOM quality from polar environments have revealed generally lowly-decomposed
organic molecules (Dziadowiec, 1994, Lupachev et al, 2017), which preserve much of the



chemical character of their precursor material due to slow progress of humification (Davidson
and Jansens, 2006). This is very important because polar soils are characterized by the specific
composition of the humification precursors.
The structure and molecular composition of the Antarctic SOM has been investigated
using $^{13}$C-NMR methods (Beyer et al, 1997, Abakumov, 2017) and it was shown that in typical
organo-mineral soils the aliphatic carbon prevails over the aromatic one, owing to the non-
ligniferous nature of its precursor material (Calace et al, 1995). Also, analyses of cryptogam
extracts were conducted towards identification of individual organic precursors (Chapman et al,
1994). This feature was then shown to be typical for soils from different regions of the Antarctic
(Abakumov, 2010), including soil formed on the penguin rookeries (Abakumov, Fattakhova,
2015). The northern most soil of Arctic polar biome shows the same trend in organic molecules
organization: higher prevalence of aliphatic structures over aromatic ones. The diversity of the
individual components in aromatic and aliphatic areas is usually higher in Arctic soil because of
the increased diversity of humification precursors (Ejarque, Abakumov, 2016, Abakumov,
2010). The over-moistened Antarctic histic soils under algae are characterized by a
predominance of proteins containing nitrogen compounds and a slight degradation of
carbohydrates in the SOM. A selective preservation of the alkyl moieties in the deeper soil layers
has been suggested, and little transformation processes of the SOM are detectable because soil
temperatures are not high enough to stimulate further microbial break-down, even in the summer
(Beyer et al, 1997). Previous reports on organic matter mainly focused on gelisols or cryosols
derived from bryophytes, algae, and vascular plants from stable habitats without pronounced
ornithogenic effects ()Carvahlo et al, 2010). It has been shown that ornitochoria play an essential
role in redistribution of plant remnants in the Antarctic (Parnikoza et al, 2016) as birds transport
considerable amounts of variably composed organic material within its inland landscapes. The
presence of organic matter of ornithogenic origin plays an important role in the formation of
humic substances. However, published data on SOM composition for the Antarctic are rare, and
further studies that detail its structural compounds and their distribution are needed. Recently,
$^{13}$C-NMR was successfully used to detail the soils found in endolitic communities in Eastern
Antarctica and revealed that endolitic organic matter is characterized by a low prevalence of
alkyl aromatic compounds (Mergelov et al, 2018).
This study aimed to compare the structural composition of the SOM from both superficial
and partially isolated (i.e. buried spots on the border with permafrost) areas and to evaluated the
stabilization rate of Antarctic cryosols. To date, this type of investigation has only been
performed on cryosols of the Kolyma lowland (Lupachev et al, 2017), where the organic matter
of modern and buried soils vary greatly in terms of their molecular composition and quality. The
objectives of the study were: (1) to evaluate the alterations in the elemental compositions of the
HAs under partial isolation (2) to assess the ratios of aromatic and aliphatic carbon species in the
topsoil and isolated areas; (3) to characterize the biochemical activity of the HAs (e.g. free
radical concentration).
**2.   Materials and Methods**
*2.1. Study sites*
King George Island is the largest in the South Shetland archipelago and only around 5%
of its 1400 km$^2$ area is free of ice (Fig. 1) (Rakusa-Suszczewski, 2002). The Fildes Peninsula and
Ardley Island, together around 33 km$^2$, comprise the largest ice-free area on King George Island
and the second largest of the South Shetland Islands. It has a gentle landscape consisting of old
coastal landforms with numerous rocky ridges and an average height of 30 m above main sea
level (AMSL) (Michel et al, 2014). According to Smellie (Smellie et al, 2014), this area mainly





consists of lava with small exposures of tuffs, volcanic sandstones, and agglomerates. The
climate is cold and humid with a mean annual air temperature of -2.2°C and mean summer air
temperatures above 0°C for only up to four months (Wen et al, 1994). The mean annual
precipitation is 350 - 500 mm/year. The Fildes Peninsula and Ardley Island are among the first
areas in maritime Antarctica to become ice-free after the last glacial maximum (Birkenmajer,
1989). The Fildes Peninsula was covered by glaciers from 8000 to 5000 BP (Mausbacher et al,
1989, Haus et al, 2014). The patterned ground in this region dates from 720 to 2640 BP. In the
South Shetland Islands, permafrost is sporadic or non-existent at altitudes below 20 m AMSL
and occurs discontinuously in altitudes from 30 to 150 m AMSL (Bockheim et al, 2013).
Mosses, lichens, and algae are common to this area along with two vascular plants (*Deschampsia
antarctica* and *Colobanthus quitensis*). Penguins, seals, and seabirds inhabit the coastal areas and
greatly impact the soil development. Major cryogenic surface-forming processes in this region
include frost creep, cryoturbation, frost heaving and sorting, gravity, and gelifluction (Michel et
al, 2014). Eight separate sites on the Fildes Peninsula have been collectively designated an
Antarctic Specially Protected Area (ASPA 125) largely due to their paleontological properties
(Management plan, 2009). The average thickness of the soil is about 15 - 25 cm. Soils from King
George Island have been divided into six groups (WRB, 2014): Leptosols, Cryosols, Fluvisols,
Regosols, Histosols, and Technosols; this corresponds well with previously published data
(Navas et al, 2008).
Three soils were selected for humic substance isolation and further investigation in this study.
All soils have top humus layers with a high carbon content and distinguishable layers of
suprapermafrost accumulation of organic matter. All three soils are classified as Turbic Cryosols
(Histic, Stagnic) (WRB. 2014). Soil profiles 1, 2, and 3 (SP1, SP2, SP3) were collected from
locations described by the following coordinates: 62,14,391 S, 58,58,549 W; 62,13,140 S,
58,46,067 W; and 62,10,578 S, 58, 51,446 W respectively. Sampling depth was 0 - 10 cm for the
superficial layers and 50 - 55, 15 - 20, 20 - 25 for SP1, SP2, and SP3 respectively. Images of the
soil profiles are presented in Fig. 2. SP1 is from under the mixed lichen-bryophyta cover, SP2
and SP3 are formed under species of *Bryophyta* and *Deshampsia antarctica* correspondingly.

### 2.2. Basic characterization

Soil samples were air-dried, ground, and passed through 2-mm sieve. Routine chemical
analyses were performed using classical methods: C and N content were determined using an
element analyzer (Euro EA3028-HT Analyser) and pH in water and in salt suspensions using a
pH-meter (pH-150 M).

### 2.3.    Extraction of humic acids (HAs)

HAs were extracted from each sample according to a published protocol (Shnitzer, 1982),
http://humic-substances.org/isolation-of-ihss-samples/). Briefly, the soil samples were treated
with 0.1 M NaOH (soil/solution mass ratio of 1:10) under nitrogen gas. After 24 hours of
shaking, the alkaline supernatant was separated from the soil residue by centrifugation at $1,516 \times$
g for 20 minutes and then acidified to pH 1 with 6 M HCl to precipitate the HAs. The
supernatant, which contained fulvic acids, was separated from the precipitate by centrifugation at
$1,516 \times$ g for 15 minutes. The HAs were then dissolved in 0.1 M NaOH and shaken for four
hours under nitrogen gas before the suspended solids were removed by centrifugation. The
resulting supernatant was acidified again with 6 M HCl to pH 1 and the HAs were again isolated
by centrifugation and demineralized by shaking overnight in 0.1 M HCl/0.3 M HF (soil/solution
ratio of 1:1). Next, the samples were repeatedly washed with deionized water until pH 3 was
reached and then finally freeze-dried. HA extraction yields were calculated as the percentage of
carbon recovered from the original soil sample (Vasilevitch et al, 2018, Abakumov et al, 2018).





### *2.4.    Characterization of humic acids (HAs)*

Isolated HAs were characterized for their elemental composition (C, N, H, and S) using the Euro EA3028-HT analyzer. Data were corrected for water and ash content. Oxygen content was calculated by difference. The elemental ratios reported in this paper are based on weight. Solid-state $^{13}C$-NMR spectra of HAs were measured with a Bruker Avance 500 NMR spectrometer (Karlsruhe, Germany) in a 4-mm ZrO2 rotor. The magic angle spinning speed was 20 kHz in all cases and the nutation frequency for cross polarization was u1/2p 1/4 62.5 kHz. Repetition delay and number of scans were 3 seconds. Groups of structural compounds were identified by their chemical shifts values: alkyl C (−10 to 45 ppm), O/N-alkyl C (45 to 110 ppm), aryl/olefine C (110 to 160 ppm), and carbonyl/carboxyl/amide C (160 to 220 ppm) (Kniker, 2007). The $^{13}C$-NMR study was also conducted in bulk soil samples towards characterizing changes in the initial soil material during humification.

The ESR spectra (only for HAs due to low ash content) were recorded on a JES FA 300 spectrometer (JEOL, Japan) in X-diapason with a free-radical modulation amplitude of 0.06 mT and a microwave power in the cavity of 1 mW. Magnesium powder with fixed radical concentration was used as an external standard. The concentration of the paramagnetic centers in powdered samples was determined by comparison to relative signal intensities of the external standard using the program JES-FA swESR v. 3.0.0.1 (JEOL, Japan). (Chukov et al, 2017).

### *2.5.    Statistics*

Statistical data analysis was performed using the STATISTICA 10.0 software (ANOVA, Statistica Base 12.6, Dell, Round Rock, TX, USA). One-way analysis of variance (ANOVA) was applied to test the statistical significance of the differences between the data, based on estimation of the significance of the average differences between three or more independent groups of data combined by one feature (factor). Fisher's Least Significance Test (LST) was used for post-hoc analysis to provide a detailed evaluation of the average differences between groups. A feature of this post-hoc test is inclusion of intra-group mean squares when assessing any pair of averages. Differences were considered significant at the 95% confidence level. Concentrations of organic and inorganic contaminants were determined with at least three replicates. The calculated average concentrations are provided as mean ± standard deviation (SD).

## 3. Results and Discussion

Total organic carbon (TOC) content was high in both the superficial and buried soil layers. This is indicative of the low degree of decomposition and transformation of the precursor material and is comparable to the data on soils from the Yamal tundra (Ejarque, Abakumov, 2016) and the Argentinian islands (Parnikoza et al, 2016). High TOC content is typical for the Antarctic Peninsula compared to soils of the Eastern Antarctic (Beyer et al, 1997, Mergelov et al, 2017). While both were elevated, the TOC was higher in the superficial levels relative to the lower ones. Previous studies describe high variability in the TOC content from the soils of King George and Galindez Islands, mainly depending on the diversity of the ecotopes and the sources of organic matter (Abakumov, 2010, Parnikoza et al, 2016). Isolated (buried) soil spots are not connected with fresh sources of organic matter, explaining why the TOC content in these layers is lower. The carbon to nitrogen ratio was narrowest in SP1, which was affected by the scuas' activity (evidenced by remnants of nests). This is in line with previous studies that documented the well-pronounced ornithogenic effects on soil's nitrogen content (Simas et al, 2007, Parnikoza



et al, 2016). Fine earth of soils investigated characterized by acid reaction, which is expected for
soils of this region.
In terms of elemental composition, soil HAs are comparable with those previously
reported for the Arctic and Antarctic soil. Current exposed organic layers contain HAs with
higher carbon and nitrogen and lower oxygen content. Conversely, the HAs of isolated soil
patches show increased levels of oxidation. In comparison to soils of the tundra in the Komi
Republic (Vasilevitch et al, 2018), HAs found in this study were more oxidized, comparable to
those of the Kolyma Lowland (Lupachev et al, 2017) and previously published data from the
Fildes Peninsula (Abakumov, 2017).
Data on the distribution of carbon species in HAs (fig. 3) and in bulk soil (fig. 4)  samples
indicated that aromatic compound content is generally lower than the alkyl components. This is a
well-known peculiarity of the soils of the polar biome (McKnihct et al, 1994, Beyeret al, 1997).
At the same time, the degree of aromaticity of the isolated HAs is three fold higher than in the
bulk organic matter. This suggests the presences of the humification process in the soils of
Antarctica since humification involves increasing the aromatic compound content in
macromolecules. This supports the classical humification hypothesis instead of new arguments,
which are critical for this approach (Lehman, Kleber, 2015). Our data shows that SOM is on a
continuum and HAs are the main acting constituent of this continuum; thereby confirming that
this model is applicable even in Antarctica. The degree of aromaticity was higher in both isolated
HAs and bulk soil samples from superficial levels compared to samples from isolated patches.
Carbonyl/carboxyl/amide area (160 - 220 ppm) was more prevalent in the HAs of topsoils and
less abundant in the organic matter of bulk samples (this region was presented mainly by
carboxylic and amid carbon in the interval between 160 - 185 ppm) (Kniker, 2007). HAs
extracted form SP1, located under the *Deshampsia antarctica,* exhibited wide peaks around 110 -
140 ppm (H-aryl, C-aryl, olefinic-C) and at 140 - 160 ppm (O-aryl and N-aryl-C), while
aromatic components of SP2 and SP3 were mainly represented by peaks between 110 - 140 ppm.
This difference can be explained by the organic remnants of *Deshampsia antarctica* serving as
the precursor for humification. All HA samples showed intensive areas of alkylic carbon (0 - 45
ppm), aliphatic C and N, and methoxyl C (45 - 110 ppm), O-alkyl of carbohydrates and alcohols
(60 - 95 ppm), and acetal and ketal carbon of carbohydrates (95 - 110 ppm). Carbon composition
of the bulk samples was different from isolated HAs as evidenced mainly by the presence of
alkyl carbon (0 - 45 ppm) and O- and N-alkyl carbon (45 - 110 ppm). Characteristic features of
the bulk organic matter include carboxylic carbon and aryl compound content was low relative to
isolated HAs. Only soils with prior ornithogenic interactions showed increases in carboxylic
peaks, which corresponds well to data on relic ornithogenic soil (Beyer et al, 1997).
The C-alkyl : O-N-alkyl ratio used to indicate the degree of organic matter transformation
was quite variable in all samples investigated. This can be caused by diversity in the origin and
composition of the humification precursors. In case of comparisons with humic and fulvic acids
of tundra soils (Vasilevitch et al, 2018), HAs of soils investigates are intermediated between
HAs and fulvic acids of tundra Histosols with partially decomposed organic matter. These data
are in line with a previous report (Hopkins et al, 2006) that showed soils of the Antarctic Dry
Valleys have low alkyl-C : O-alkyl-C ratio using solid-state [13]C-NMR spectroscopy) and,





therefore can serve as a labile, high-quality resource for micro-organisms. Beyer et al (1997) showed that both the CPMAS [13]C-NMR and the Py-FIMS spectra of the Terri-Gelic Histosol were dominated by signals from carbohydrates and alkylic compounds, which is corroborated by our findings. They also suggest that the [13]C-NMR data reflected decomposition of carbohydrates and enrichment of alkyl-C in deeper soil layers. In regards to the bulk SOM, this was true for SP2 and SP3 but not for SP1 that formed under the vascular plant *Deshampsia Antarctica*.

A representative electrone spin resonanse ESR spectrum of HAs is presented in fig 5 and the ESR parameters are similar to HAs and FAs of temperate soils (Senesi, 1990, Senesi et al, 2003). The spectra show a single, wide line with a g-factor ranging from 1,98890 to 1,99999, attributable to the presence of stable semiquinone free radicals in the HA-containing macromolecules (Table 5). The free radical content was higher in the superficial levels than in the isolated ones. This corresponds well with previous reports (Chukov et al, 2017, Abakumov et al, 2015) that connect the isolation of buried organic matter in the supra-permafrost with declining free radical content. This reveals the increased biochemical activity of HAs in topsoil. Compared to data from Lupachev (2017), the differences between exposed and isolated areas are less pronounced but, in general, the HAs of the Antarctic soils contain more unstable free radicals on average than the tundra soils of the Kolyma Lowland (Lupachev et al, 2017) and are comparable to the soils from the Yamal tundra (Chukov et al, 2017). Taken together, the free radical content found in our study was lower than in anthropogenically affected boreal and forest steppe soils of the East-European plains (Abakumov et al, 2018).

## 4. Conclusions

High TOC content was fixed for the three studies representatives of Turbic Cryosols on King George Island, Northwest of the Antarctic Peninsula, Western Antarctic. High amounts of TOC are characteristic for both superficial and partially isolated soil materials. HAs contained three fold more aromatic carbon than bulk SOM, which indicates that humification appears and is active in soils of the Antarctic. Moreover, the amounts of aromatic carbon and carboxyl groups were higher in the HAs of the superficial layer, which is likely caused by the greater diversity of their organic precursors and more active humification than in sub-aerial conditions. The HAs of the superficial sample layers contained lower concentrations of free radicals, an indicator of active transformation in the topsoil. In general, the organic matter from partially isolated areas is less stable in terms of carbon species and free radical content. This likely results from the relative lack of fresh organic precursors and the different aeration and hydration conditions of stagnification bordering the permafrost table.

**Acknowledgments:** This work was supported by the Russian Foundation for Basic Research, project No 16-34-60010 and 18-04-00677a and the Saint Petersburg State University Internal Grant for the Modernization of Scientific Equipment No. 1.40.541.2017. Analyses were carried out at the Center for Magnetic Resonance and at the Center for Chemical Analysis and Materials Research of Research Park of St. Petersburg State University, Russia.

The authors would like to thank Dr. A. Lupachev for assistance with field research and providing images for Figure 2 (2-1, 2-2 and 2-3).





310

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





Table. 1. Basic characteristics of soils

| Sample | TOC, % | N, % | C/N | $pH_{H2O}$ | $pH_{CaCL2}$ | Color |
|---|---|---|---|---|---|---|
| 1  O | 27.63±0.23 | 5.18±0.42 | 5.33 | 6.35 | 5.30 | 10 YR 4/7 |
| 2  [CRH] | 19.05±0.15 | 2.20±0.05 | 8.66 | 5.67 | 4.89 | 2.5 YR 4/4 |
| 3  O | 20.04±0.17 | 1.16±0.09 | 17.13 | 4.80 | 4.80 | 10 YR 4/4 |
| 5  [CRH] | 12.33±0.24 | 0.78±0.09 | 15.80 | 4.70 | 4.50 | 2.5 YR 4/3 |
| 4  O | 10.16±0.09 | 0.84±0.07 | 11.98 | 4.90 | 4.21 | 10 YR 5/3 |
| 6  [CRH] | 6.66±0.07 | 0.81±0.09 | 8.20 | 4.70 | 4.35 | 2.5 YR 5/3 |








Table 2. Elemental composition (%) and atomic ratios in HAs

| Sample № | C | N | H | O | C/N | H/C | O/C |
|---|---|---|---|---|---|---|---|
| 1 | 49.53±0.56 | 5.55±0.07 | 6.90±0.11 | 38.02±0.64 | 8.92 | 0.13 | 0.76 |
| 2 | 47.14±0.45 | 4.30±0.06 | 6.79±0.09 | 41.77±0.21 | 10.96 | 0.14 | 0.88 |
| 3 | 45.55±0.32 | 5.14±0.09 | 5.80±0.09 | 43.51±0.35 | 8.86 | 0.12 | 0.95 |
| 4 | 43.77±0.24 | 4.72±0.11 | 6.90±0.08 | 44.61±0.21 | 9.27 | 0.15 | 1.01 |
| 5 | 49.99±0.41 | 4.78±0.08 | 6.56±0.08 | 38.67±0.34 | 10.45 | 0.13 | 0.77 |
| 6 | 44.45±0.034 | 3.99±0.07 | 6.77±0.10 | 44.79±0.25 | 11.14 | 0.15 | 1.01 |
| P, One way Anova, superficial/buried | 0.14 | **0.05** | 0.29 | **0.05** | n.d. | n.d. | n.d. |






Table 3. Carbon species integration in molecules of the HAs, %

| Sample № | Carbonyl/ carboxyl/ amide | Aryl-olefine | O-N alkyl | Calkyl | Calkyl/O-N alkyl | Caryl/Calkyl |
|---|---|---|---|---|---|---|
| | 220-160 | 160-110 | 110-45 | 45-0 | | |
| 1 | 11,38 | 33,59 | 39,86 | 14,18 | 0.35 | 2.36 |
| 2 | 10,75 | 30,45 | 31,86 | 26,05 | 0.81 | 1.16 |
| 3 | 19,24 | 23,34 | 29,54 | 27,85 | 0.94 | 0.83 |
| 4 | 16,48 | 21,42 | 34,23 | 27,87 | 0.81 | 0.77 |
| 5 | 16,75 | 33,40 | 29,12 | 20,71 | 0.71 | 1.61 |
| 6 | 14.39 | 26.86 | 40.07 | 18.68 | 0.46 | 1.43 |
| P, One way Anova, superficial/buried | **0.02** | **0.03** | **0.02** | 0.73 | n.d. | n.d. |





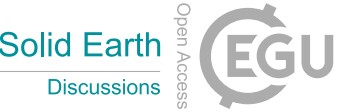


Table 4. Carbon species integration in molecules of the bulk organic matter, %

| Sample № | Carbonyl/ carboxyl/ amide | Aryl- olefine | O-N alkyl | Calkyl | Calkyl/O -N alkyl | Caryl/Calkyl |
|---|---|---|---|---|---|---|
| | 220-160 | 160-110 | 110-45 | 45-0 | | |
| 1 =113=O | 7.24 | 11.37 | 46.20 | 35.19 | 0.76 | 0.32 |
| 2 113-Ch | 18.23 | 10.29 | 40.59 | 30.89 | 0.76 | 0.33 |
| 3 123 O | 7.34 | 20.48 | 55.12 | 17.06 | 0.31 | 1.20 |
| 4 123 Ch | 9.34 | 11.27 | 49.50 | 29.90 | 0.60 | 0.37 |
| 6 149 O | 5.72 | 13.84 | 62.22 | 18.22 | 0.29 | 0.75 |
| 6 149 Ch | 22.95 | 9.89 | 46.92 | 20.24 | 0.43 | 0.48 |
| P, One way Anova, superficial/buried | 0.53 | **0.01** | **0.05** | 0.56 | n.d. | n.d. |








Table. 5. Mass concentration of free radical in humic acids

| Soil horizon | Mass concentration of free radical, $10^{15}$ spin*g$^{-1}$ | g-factor |
|---|---|---|
| 1 | 3.67 | 2.0314 |
| 2 | 3.04 | 2.3150 |
| 3 | 3.51 | 2.0314 |
| 4 | 2.13 | 2.0303 |
| 5 | 6.10 | 2.0310 |
| 6 | 5.86 | 2.0314 |











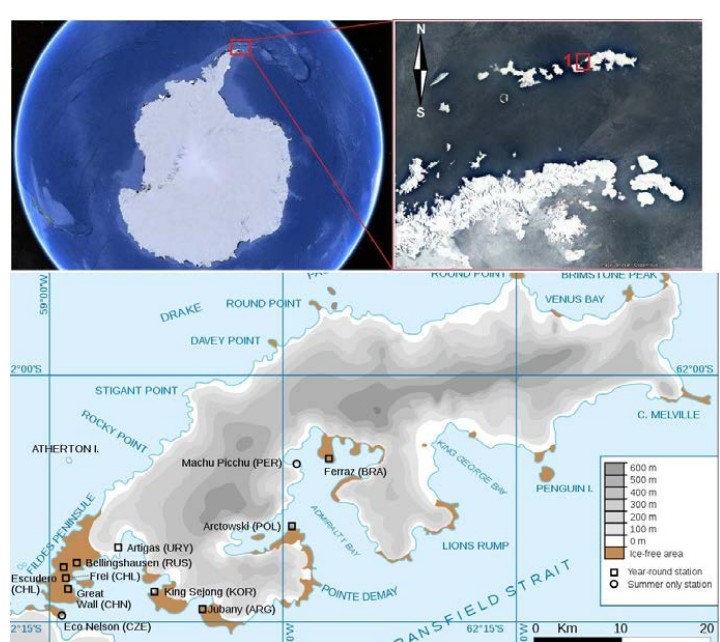


Fig. 1.

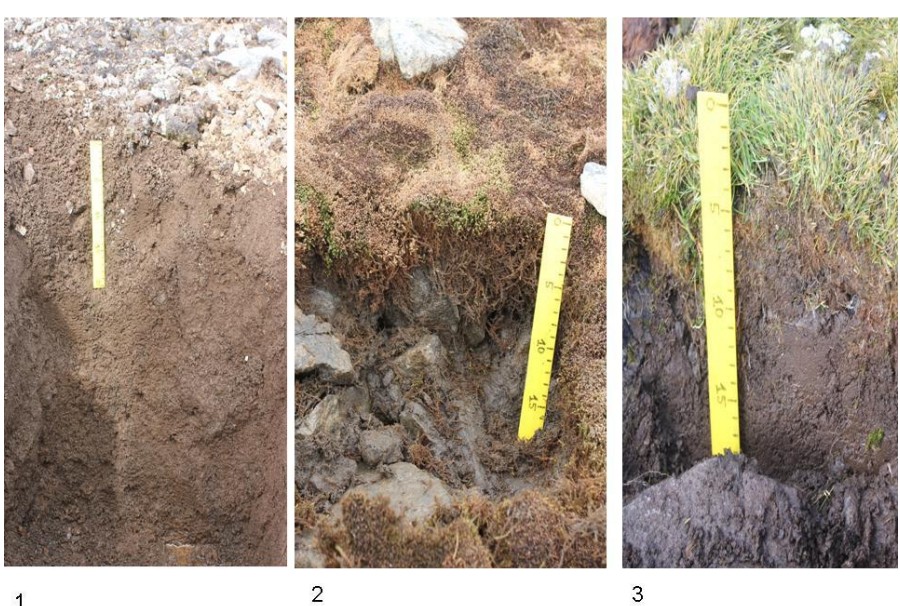

477     1                              2                              3



Fig. 2

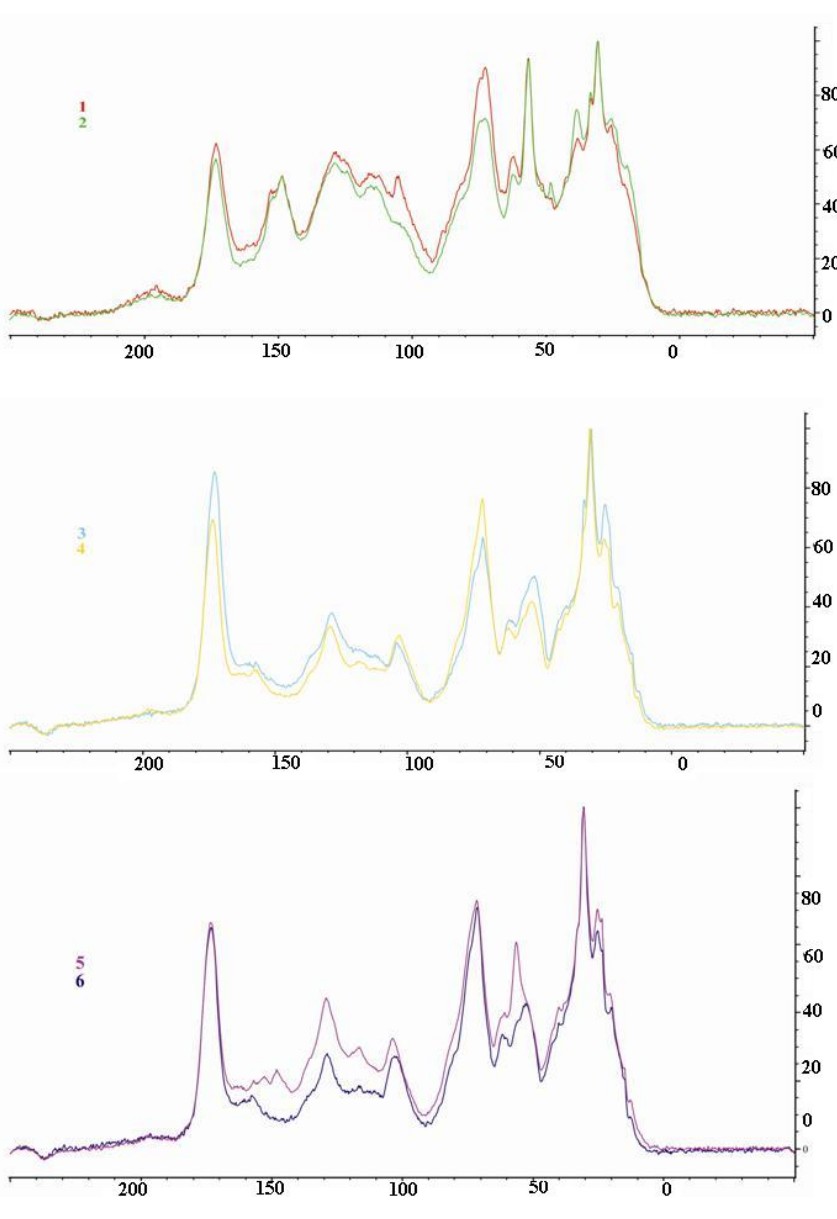



Fig. 3

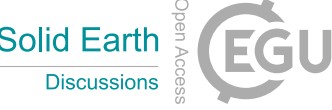

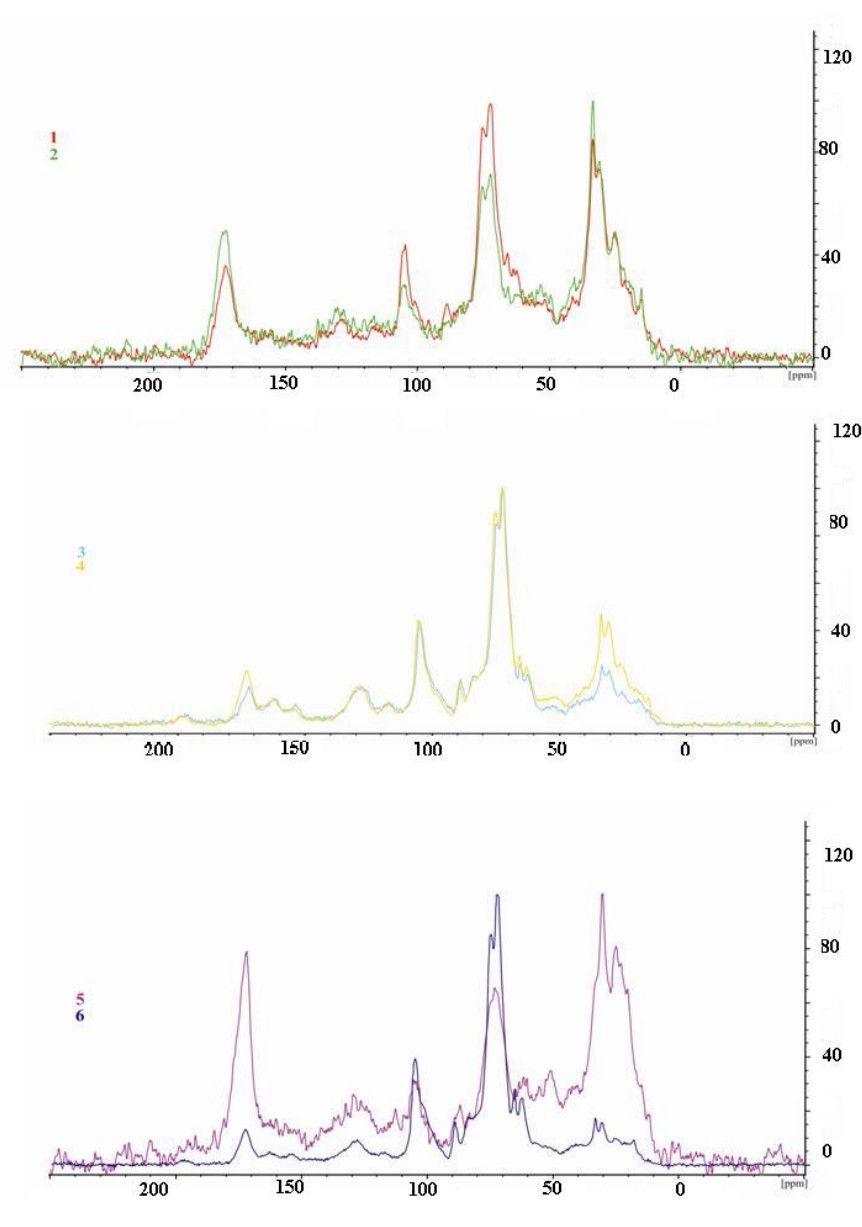



Fig. 4



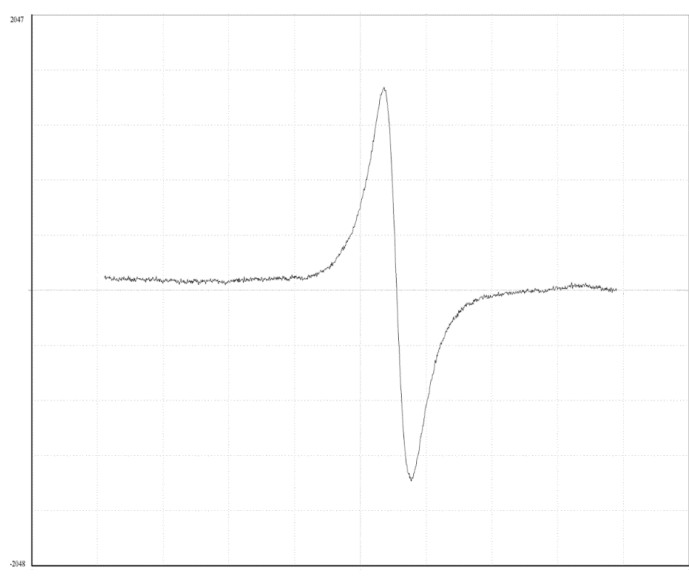


Fig. 5.


Fig. 1. Location of the Fildes peninsula
Fig.2. Soil morphology
Figure 3. 13-C NMR spectras of the HAs, isolated from soils (1-6 – according table 1)
Fugure 4. 13-C NMR spectras of bulk organic matter of soils ((1-6 – according table 1)
Figure 5. Typical ESR spectrum of humic substances investigated
Table. 1. Basic characteristics of soils
Table 2. Elemental composition (%) and atomic ratios in HAs
Table 3. Carbon species integration in molecules of the HAs, %
Table 4. Carbon species integration in molecules of the bulk organic matter, %
Table. 5. Mass concentration of free radical in humic acids