# Peer review of "STABILITY OF SOIL ORGANIC MATTER IN CRYOSOLS OF MARITIME"

_Solid Earth, 2018_

## Short Comment (SC1) · 3 Jun 2018

Overall, this article showed chemical constructions in humic acids from three different soil samples in different conditions in Antarctica. The authors pointed out the pronounced differences of free radicals and aromatic carbon between superficial layer and isolated soil materials. This was explained by the process of humification and comparing carbon species, which is quite interesting topic. However, this article needs a revision of the language. For example, there were several incorrect tense and too long sentences which are too difficult to understand. Besides, one paragraph (Page

6, line 243-259) is too long (394 words) so that it's hard to follow. Each paragraph should keep one simple idea. Table1-4 isn't shown in the article. In Table 1, O and CRH should be elaborated. In Table 3, what the second row stands for? The same for table 4. Please, add the title to each figure. From figure 3-5, there are even no x and y coordinates. Some other minor errors and questions are as below (not all):

Line 108, (Cravahlo et al., 2010) Line 114, "endolitic", is it "endolithic"? Line 156, the information of soil profile is better shown in form of table. Line 218, "describe", "described" Line 223, "is", was Line 235, "aromatic . . . . . .", aromatic compound content was generally lower than the alkyl components Line 240, what argument the authors mentioned here? The carbon spices in humification? Line 241, "shows", showed Line 260 and 266, alkyl-C and C-alkyl, are they the same? Line 268, the whole name of CAPMAS and Py-FIMS Line 270, "suggest", suggested Line 272, "that formed"ïïjŇthat was formed Line 275, "show", showed Line 279. "connect", connected Line 281-284, please rephrase this sentence. Too long to follow. Line 290-291, "Has. . . . . .bulk SOM", HAs contained three-fold more aromatic carbon than that in bulk SOM

---

## Referee Comment (RC1) · Anonymous Referee #1 · 26 Jun 2018

STABILITY OF SOIL ORGANIC MATTER IN CRYOSOLS OF MARITIME 2 ANTARCTIC: INSIGHTS FROM 13-C NMR AND ELECTRON SPIN RESONANCE 3 SPECTROSCOPY

Evgeny Abakumov, Ivan Alekseev

row 108 delete ) rows 138, 139, 140, 322, 323 - letter size is different row 375, affect, error - affecr 434 intervals are needed for the initials of the names row 451, Table 1 - The organic carbon content is two high (from over 6 to over 25%) There are investigations in Livingston island, where the organic carbon content is low even around pinguinum rockeries

I ecomend the following papers:

Sokolovska, M., L. Petrova, N. Chipev. Particulars of Humus Formation in Antarctic Soils: Factors, Mechanisms, Properties. – Bulgarian Antarctic Research: Life Sciences, 1, 1996. - 7–12. Sokolovska, M., N. Chipev, R. Ilieva, M. Nustorova, L. Petrova, Z. Vergilov, R. Hristova, J. Bech. Soils in Livingston Island: Composition, properties and ecological aspects. - In: Bulgarian Antarctic Research, A Syntesis (Ed. Pimpirev, C. and N. Chipev), Sofia, "St. Kl. Ohridski" University Press, 2015. – 308-319.

---

## Referee Comment (RC2) · Anonymous Referee #2 · 5 Jul 2018

The manuscript, titled "Stability of soil organic matter in Cryosols of Maritime Antarctic: Insights from 13-C NMR and electron spin resonance spectroscopy" is devoted to problem of soil organic matter stabilization in Cryosols of Maritime Antarctic with special reference to molecular structure and biochemical activity of the molecules.

Following comments are made with aim to improve the quality of manuscript: 1. list of references should be improved and styled according to the journal requirements. 2. table 2 – clarify if data presented in mass or atomic values? 3. table 5 better to call "Free radical concentration in….." and improve the second collumn. 4. table 5 – clarify

the error of ESR methods 5. sample codes in table 4 do not correspond to table 1. 6. fig 1 – It is not evident were sampling point were located. 7. fig 2 – capture – clarify 1,2 and 3 designations. 8. fig 3-5 are low quality and axes are not designated.

Other minor issues: Line 13-15, 34-36 – key words are the same as highlights Line 80 Lodigin change to Lodygin Line 80 – excess braket Line 186 - correct "ZrO2". Line 189 , table 3, 4 - aryl/olefin change to aromatic Table 3 and 4 Calkyl, Calkyl/O-N alkyl, Caryl/Calkyl change to Alkyl, alkyl/O-N alkyl, Arom/Alkyl Table 3 and 4 – lines 2 and 5 – add "Chemical shift, ppm" Line 487 Figure 4. Change to Fig. 3. Line 488 – Figure 5. Change to Fig. 4.

---

## Short Comment (SC2) · 5 Jul 2018

The article presents interesting and needed issues related to coal and organic matter in the soil of Cryosols of Maritime Antarctic.

The methodology is a bit unclear. First: name of the text. Why those specific depths have to be chosen for different soil profiles (50 - 55, 15 - 20, 20 - 25 for SP1, SP2, and SP3 respectively). In tab. 1 we've got 1O, 2 [CRH ... in tab 2 we've got only sample numbers and in tab 4 - other names. Placing soil samples on the map would

be convenience. Results should be given up to 3 significant figures (there are 3 and 4). There are no figures names while tables are named properly.

The pH in sample 1 is quite high. Was it also an ornithogenic effect?

---

## Author Comment (AC1) · 6 Jul 2018

Dear reviwer! thank you very much for suggestions for reorganization of the manuscript text. We will make sentences shorter. We will lso clarify information about relationship between humicication rate and carbon species contnet. Thank you alsos for suggestion related to grammar issueses.
* * *

---

## Short Comment (SC3) · 7 Jul 2018

Dear reviewer, Thank you very much for your advices for improving our manuscript - We will clarify methodoly part more detailed - Tables and their content will be reviewed and improved as well as figures - Discussion on soil parameters will be extended

---

## Short Comment (SC4) · 7 Jul 2018

Dear reviewer, Thank you very much for taking time for review of our manuscript

- We will check and correct rows you mentioned - Organic matter contents will be discussed more detailed in the text of manuscript - Thank you for suggested literature, we will review these articles

---

## Short Comment (SC5) · 7 Jul 2018

Dear reviewer,

Thank you very much for taking time to review our manuscript

- Tables and figures content will be reviewed, improved - We will also clarify the points you mentioned - Reference list will be verified with journal's requirements - Your suggestions for correcting spelling and logical errors will be used

---

## Author Comment (AC2) · 25 Jul 2018

Dear Referees,

Thank you very much for your advices, which are crucial for improvement of our article.

We have modified our manuscript with your remarks as following: - All the spelling errors have been checked, incorrect spellings were corrected (Line 80, 114, 186, 279, 375); - Paragraphs and sentences have been rephrased when it was hard to follow (lines 281-284, 290-291); - Abbreviations of carbon species have been care-

fully checked and corrected (line 260, 266); - Tables 1-4 have been referenced in the text of our manuscript; - X and Y axis have been captured in figures 3-5; - Letter size has been checked and corrected in mentioned lines (lines 138, 139, 140, 322, 323); - Missing intervals in the reference list have been added; - The data from Table 2 have been clarified and represent mass values; - Names of carbon species have been changed accordingly in Tables 3,4; - The name of Table 5 has been changed; - Key points/Highlights sections have been reviewed and improved; - Reference list has been checked and improved according to the journal's requirements; - The figure with sampling plots and study area has been carefully checked and improved, insert map of Fildes peninsula with sampling plots has been added; - Soil samples names mentioned in tables 1-5 has been corrected to the right ones (1 O, 1 [CRH]; 2 O, 2 [CRH], 3 O, 3 [CRH])

---

## Referee Report (RR1)

[referee-annotated manuscript omitted]

---

## Author Response (AR2)

Dear reviewer!

Thank you very much for all remarks and suggestions.

The manuscript text was improved according all remarks.

Also additional reference was given regarding the occurrence of Technosols in Antarctica (this could be especially possible in surroundings of Bellinshausen station)

With Kind Regards,

Evgeny Abakumov, corresponding author

[revised manuscript text omitted]

**Comment [АИИ7]:** This reference has been added to prove the inclusion of Technosols

**Comment [АИИ8]:** The chapter has been renamed according to reviewers' comment 
[revised manuscript text omitted]

---

## Author Response (AR3)

Dear reviewer,
Thank you very much for taking time to review our manuscript and your vital advices on
improving the text. We have corrected our manuscript according to your remarks

| Remark | Response |
|---|---|
| l. 46 continuous and discontinuous permafrost result in the stabilization of essential…
 Is it really continuous/discontinuous…) or active layer dynamics that drives these processes? | We have changed the term in mentioned sentence |
| l. 129 be careful, what is the meaning of an average altitude in the area?
 Fildes consists of two main plateaus separated by a lower relief. I don't understand why you indicate the mean altitude of the ice-free area. It is not representative of the region | We have deleted the information on the mean altitude |
| | The term has been re-written in a right form |
| | The wrong information has been corrected |
| . 135 Last Glacial Maximum | The line mentioned in remark has been checked and corrected |
| l. 136 this is not true. The onset of deglaciation started as in many other areas in the SSI by 8000-9000 ka and spread during the mid Holocene. Check Oliva et al (2016) in Geomorphology, from Byers, and references about other areas in the SSI therein.
 Oliva, M.; Antoniades, D.; Giralt, S.; Granados, I.; Pla-Rabes, S; Toro, M.; Sanjurjo, J.; Liu, E.J. & Vieira, G. (2016). The Holocene deglaciation of the Byers Peninsula (Livingston Island, Antarctica) based on the dating of lake sedimentary records. Geomorphology, 261: 89-102. | Results and Discussion chapter has been extended with comparison of soils studied in our work  and soils previously investigated in the region of Maritime Antarctica

[revised manuscript text omitted]